

# Wnt signaling in liver disease: emerging trends from a bibliometric perspective

Guangyi Jiang[1,2], Chiung-Kuei Huang[3], Xinjie Zhang[1,2], Xingyu Lv[1,2], Yifan Wang[1,2], Tunan Yu[1,2] and Xiujun Cai[1,2]

[1] Department of General Surgery, Sir Run Run Shaw Hospital, College of Medicine, Zhejiang University, Hangzhou, Zhejiang, China
[2] Key Laboratory of Laparoscopic Technology of Zhejiang Province, Sir Run Run Shaw Hospital, College of Medicine, Zhejiang University, Hangzhou, Zhejiang, China
[3] Liver Research Center, Rhode Island Hospital and The Warren Alpert Medical School of Brown University, Providence, RI, USA

Corresponding authors
Tunan Yu, 3314006@zju.edu.cn
Xiujun Cai, srrsh_cxj@zju.edu.cn

## ABSTRACT

**Background:** The Wnt signaling pathway, an evolutionarily conserved molecular transduction cascade, has been identified as playing a pivotal role in various physiological and pathological processes of the liver, including homeostasis, regeneration, cirrhosis, and hepatocellular carcinoma (HCC). In this study, we aimed to use a bibliometric method to evaluate the emerging trends on Wnt signaling in liver diseases.
**Methods:** Articles were retrieved from the Web of Science Core Collection. We used a bibliometric software, CiteSpace V 5.3.R4, to analyze the active countries or institutions in the research field, the landmark manuscripts, important subtopics, and evolution of scientific ideas.
**Results:** In total, 1,768 manuscripts were published, and each was cited 33.12 times on average. The U.S. published most of the articles, and the most active center was the University of Pittsburgh. The top 5 landmark papers were identified by four bibliometric indexes including citation, burstness, centrality, and usage 2013. The clustering process divided the whole area into nine research subtopics, and the two major important subtopics were "liver zonation" and "HCC." Using the "Part-of-Speech" technique, 1,743 terms representing scientific ideas were identified. After 2008, the bursting phrases were "liver development," "progenitor cells," "hepatic stellate cells," "liver regeneration," "liver fibrosis," "epithelial-mesenchymal transition," and etc.
**Conclusion:** Using bibliometric methods, we quantitatively summarized the advancements and emerging trends in Wnt signaling in liver diseases. These bibliometric findings may pioneer the future direction of this field in the next few years, and further studies are needed.

## INTRODUCTION

The Wnt signaling pathway, an evolutionarily conserved molecular transduction cascade, has been identified to play a pivotal role in various diseases (*Nusse & Clevers, 2017*), including degenerative diseases (*Gong et al., 2001*; *Cisternas et al., 2019*), chronic

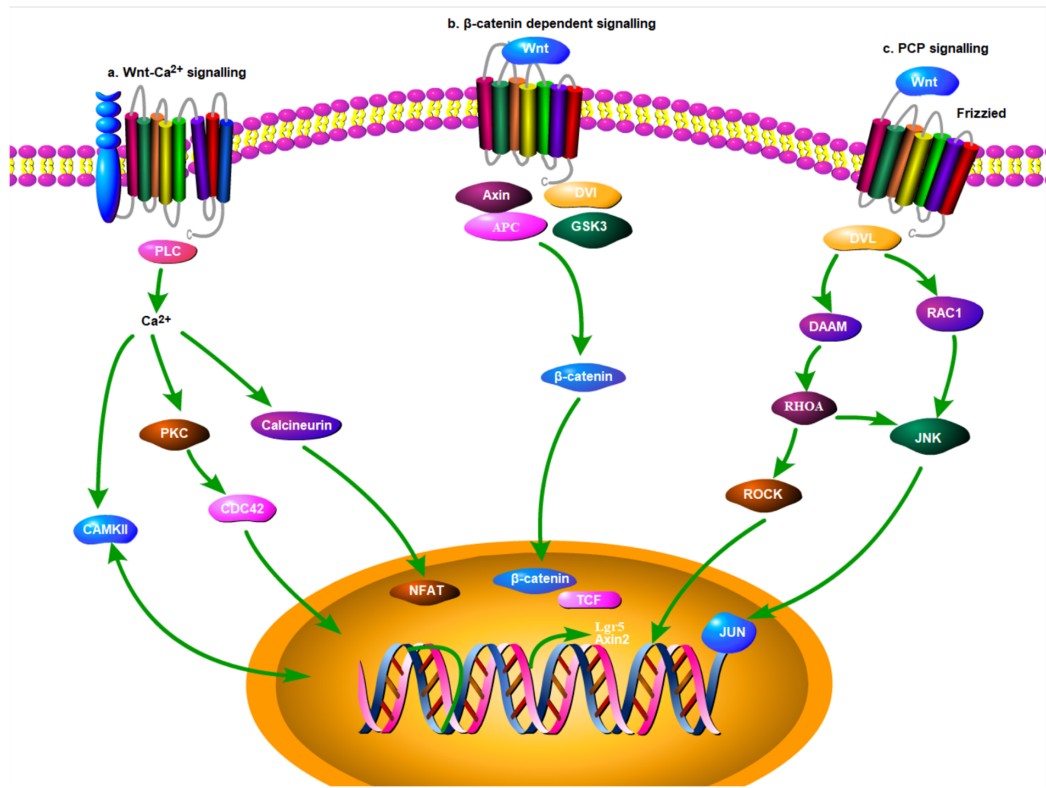

**Figure 1 Signaling pathway for Wnt molecules (Depicted using Portable Pathway Builder 2.0 from ProteinLoung).** Three classic signalling pathways were included: (A) Wnt-Ca2+ signalling; (B) β-catenin dependent signalling; (C) PCP signalling.

inflammation (*Nalesso et al., 2017*), and malignancy (*Anastas & Moon, 2013*) (Fig. 1). Dysregulation of the Wnt pathway is correlated with dysfunction in different organs including the teeth (*Yamashiro et al., 2010*), eye (*Cavodeassi et al., 2005*), and heart (*Heallen et al., 2011*). In the liver, the Wnt pathway is essential for homeostasis, embryogenesis, development, maturation, and regeneration (*Perugorria et al., 2018*). It is also associated with several pathological conditions such as cirrhosis (*Nishikawa, Osawa & Kimura, 2018*), hepatoblastoma (*Czauderna & Garnier, 2018*; *Bell et al., 2017*), and hepatocellular carcinoma (HCC) (*Wands & Kim, 2014*). Currently, several Wnt molecules with potential pharmacological value are being explored for future therapeutic interventions (*Kahn, 2014*; *Pez et al., 2013*).

Use of bibliometric methods is a novel way to summarize the advancements in a research area. Several bibliometric methods have been developed to construct knowledge maps which can detect hotspots or even emerging trends in a research area. Several softwares are avaliable to conduct research for this aim, including CiteSpace (*Synnestvedt, Chen & Holmes, 2005*), CitNetExplorer, Vosviewer (*Van Eck & Waltman, 2017*), and HistCite (*Garfield, 2004*). Although most of the bibliometric studies are currently performed in the area of scientometrics or social science (*Jia, 2017*; *Liu, 2013*), some researchers have made similar attempts in the field of biomedicine. For example, *Shuaib et al. (2015)* investigated the top 100 papers in the area of cardiovascular

research, and *Zhou & Zhao (2015)* critically analyzed studies on liposomes from 1995 to 2014. In this study, we aimed to use bibliometric methods to analyze manuscripts on the Wnt signaling pathway in liver diseases. Using CiteSpace, we aimed to identify the active countries or institutions in the research field, the landmark manuscripts, research subtopics, and the evolution of scientific ideas.

## METHODS

The authors conducted a search of literature on the Web of Science Core Collection (WOSCC) on November 15th, 2018 to obtain articles on the Wnt signaling pathway in liver diseases. The key words "Wnt" and "liver" were used. According to Prof. Chaomei Chen (*Chen, 2018*), the inventor of CiteSpace, it is unnecessary to endlessly refine search queries to eliminate papers of irrelevant topics. Instead, CiteSpace can differentiate those papers during the process of clustering. Each manuscript was downloaded in the manner of "full records and cited references." The information on the titles, keywords, author, institution, country, abstract, and references was all stored.

The bibliometric software CiteSpace V 5.3.R (64 bits) (*Synnestvedt, Chen & Holmes, 2005*) was utilized for this study. In a co-citation network for literature, every node represents a reference paper, and a link between two nodes represents the relation of citing. Co-citation maps could also be constructed for countries or institutions, and the links between two nodes represent co-operation between two countries or institutions. To simplify the structure of the co-citation network, a restriction with g-index (*Egghe, 2006*) was utilized, and the scale factor $k$ was set as 5. To evaluate the importance of a node in a network, four indexes, that is, total citation number, betweenness centrality, burstness, and usage 2013 were utilized. Betweenness centrality (*Chen, 2005*) is defined as a metric of a node in a network that measures how likely an arbitrary shortest path in the network will go through the node. The concept of "centrality" arose from studies in the social network. A node with higher centrality is more likely to connect several areas, and even be a turning point in a science domain (*Chen, 2004*). Burstness is defined as the count of citation for a node that is sharply increasing throughout time (*Kleinberg, 2003*). Usage 2013 is an index developed by the WOSCC and indicates the times of "usage" since the year 2013 by all Web of Science (WOS) users, such as by clicking links to the full-length article or saving the article for use in a bibliographic management tool (*Wang, Fang & Sun, 2016*).

Several other bibliometric techniques were also utilized. Firstly, the process of "clustering" was used to identify different research subtopics in all the papers on "Wnt signaling in liver diseases." The quality of clustering was measured by two indexes, modularity and silhouette scores (*Chen, Paul & O'Keefe, 2010*). The label of each cluster was summarized from the title of the references, with the method of log-likelihood. Secondly, changes and evolution of scientific concepts over the years were evaluated. "Circles visualization" (Fig. S1) generated by Carrot 2 (*Osiński & Weiss, 2005*) was utilized to identify keywords of importance, and give relative impact to each keyword based on its calculated value Moreover, the "Part-of-Speech" technique (*Toutanova et al., 2003*) of CiteSpace was used to retrieve noun phrases from titles. The noun phrases could be considered as substitutions of scientific ideas, and were used to construct a co-citation network. For each

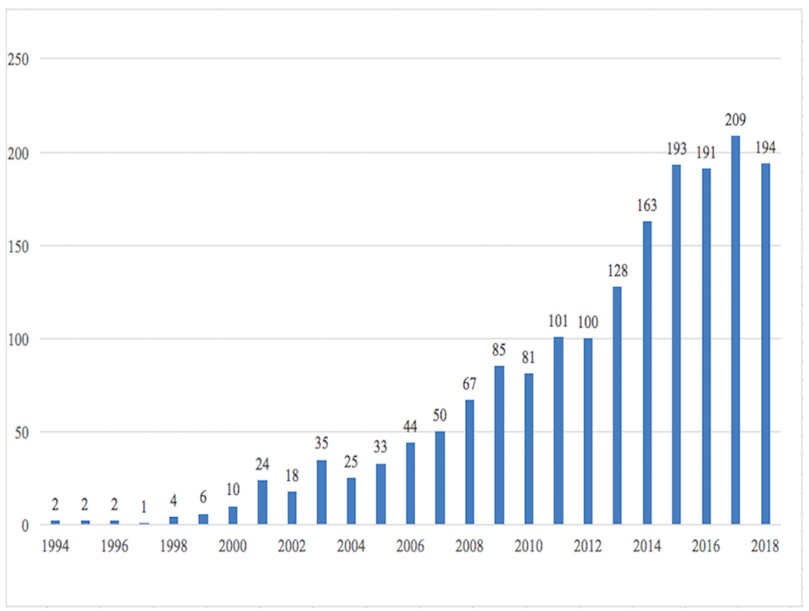

**Figure 2 Annual publications about the Wnt signaling pathway in liver diseases.**

noun phrase, the minimum number of words was two and the maximum number was four. The noun phrases with the highest burstness in the network were identified to represent the scientific ideas over the years.

## RESULTS

### Publication outputs

Using the aforementioned search strategy, a total of 1,768 manuscripts were identified. On an average, each manuscript was cited 33.12 times. The number of manuscripts increased since 1994 (Fig. 2). Among these manuscripts, there were 1,425 articles, 231 reviews, 82 meeting abstracts, 21 editorial materials, and 18 proceeding papers. Of these 1,753 manuscripts were written in English.

### Country and institutional analysis

A total of 35 countries and areas published manuscripts, and the top five countries were the U.S. (648), China (516), Germany (179), Japan (175), and France (124). Globally, a total of 1,887 institutions published manuscripts independently or cooperatively, and most of them were scattered in North America, East Asia, and Europe. The top 5 institutions were the University of Pittsburgh (99), Institut National de la Santé et de la Recherche Médicale (96), Université Paris Descartes (51), Shanghai Jiao Tong University (47), and Fudan University (43).

### Landmark manuscripts

The whole science domain was constructed with 1,768 manuscripts combined with their 57,478 references. After simplifying the network with the *g*-index, the five most cited manuscripts were identified, as the studies by *Satoh et al. (2000)*, *Benhamouche et al. (2006a)*,

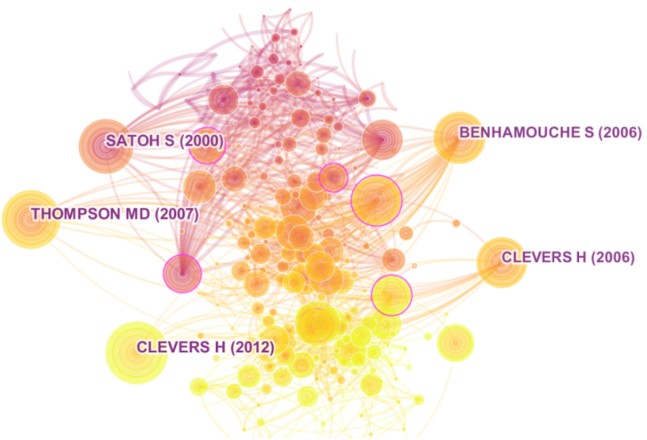

**Figure 3 Top 5 manuscripts with highest co-citation in network.**

*Thompson & Monga (2010)*, and two studies by *Clevers (2006)* and *Clevers & Nusse (2012)* (Fig. 3). Further, we selected the top 5 manuscripts by the indexes of betweenness citation, centrality, burstness, and usage 2013 (Table 1).

## Research subtopics

In Fig. 4, the whole co-citation map of manuscripts was divided into nine clusters as "liver zonation," "HCC," "human colorectal cancer," "liver tumor," "stem cell niche," "murine hepatic transit," "liver stem cell," "alpha-wnt10b signaling loop," and "endothelial cell niche." The modularity for this clustering was 0.6406, and the mean silhouette was 0.7945. Among them, the two largest clusters were "liver zonation (0#)"and "HCC (1#). In the cluster of "liver zonation," three papers citing most of the members in this cluster were the works of *Torre, Perret & Colnot (2010)*, *Gebhardt & Hovhannisyan (2010)*, and *Braeuning et al. (2010)*. In the cluster of "HCC," three papers citing most of the members in this cluster were the studies by *Nejak-Bowen & Monga (2011)*, *Monga (2011)*, and *Wei et al. (2000)*.

## Evolution of scientific ideas

Circles visualization generated by Carrot 2 (Fig. S1) was generated to identify keywords in the whole science domain. Some keywords with research importance were identified, such as "Cancer genes," "Signaling in Hepatocytes," "Pathway Mutations," "Beta-catenin in Mice," "Liver of Patients," and "Liver Stem." To a certain degree, these keywords could reflect current trends in the research area (Fig. 5A).

Using the "Part-of-Speech" technique of CiteSpace, 1,743 noun phrases in the titles of manuscripts were detected by natural language analysis. A co-citation network of the noun phrases was constructed, and 43 of the phrases were found to have burstness (Fig. S2). Among them, 15 with scientific importance were specifically analyzed (Table 2). Before 2008, the keywords of scientific importance were "beta-catenin gene," "nuclear accumulation," "adenomatous polyposis coli," "genetic alterations," "beta-catenin mutations," "HCCs," "Wnt pathway," and "liver development." After 2008, the bursting

**Table 1** Summarization of top 5 studies by the index of citation, centrality, burstness and usage 2013.

| Year | First author | Journal | Title | Top 5 citation | Top 5 centrality | Top 5 burstness | Top 5 usage 2013 |
|------|-------------|---------|-------|----------------|------------------|-----------------|------------------|
| 1998 | de La Coste A | PNAS | Somatic mutations of the β-catenin gene are frequent in mouse and human hepatocellular carcinomas (*De La Coste et al., 1998*) | | | Y | |
| 1998 | He T C | Science | Identification of c-MYC as a target of the APC pathway (*He et al., 1998*) | | Y | Y | |
| 1999 | Tetsu O | Nature | β-Catenin regulates expression of cyclin D1 in colon carcinoma cells (*Tetsu & McCormick, 1999*) | | | Y | |
| 2000 | Satoh S | Nature genetics | AXIN1 mutations in hepatocellular carcinomas, and growth suppression in cancer cells by virus-mediated transfer of AXIN1 (*Satoh et al., 2000*) | Y | | Y | |
| 2006 | Clevers H | Cell | Wnt/β-catenin signaling in development and disease (*Clevers, 2006*) | Y | | | |
| 2002 | Taniguchi K | Oncogene | Mutational spectrum of β-catenin, AXIN1, and AXIN2 in hepatocellular carcinomas and hepatoblastomas (*Taniguchi et al., 2002*) | | Y | | |
| 2006 | Benhamouche S | Developmental cell | Apc tumor suppressor gene is the "zonation-keeper" of mouse liver (*Benhamouche et al., 2006b*) | Y | | | |
| 2006 | Tan X | Gastroenterology | Conditional deletion of β-catenin reveals its role in liver growth and regeneration (*Tan et al., 2006*) | | Y | | |
| 2010 | Thompson M D | Hepatology | Wnt/β-catenin signaling in liver health and disease (*Thompson & Monga, 2010*) | Y | | | |
| 2007 | Boyault S | Hepatology | Transcriptome classification of HCC is related to gene alterations and to new therapeutic targets (*Sandrine et al., 2010*) | | Y | | |
| 2012 | Guichard C | Nature genetics | Integrated analysis of somatic mutations and focal copy-number changes identifies key genes and pathways in hepatocellular carcinoma (*Guichard et al., 2012*) | | Y | | |
| 2012 | Clevers H | Cell | Wnt/β-catenin signaling and disease (*Clevers & Nusse, 2012*) | Y | | | |
| 2012 | Li V S W | Cell | Wnt signaling through inhibition of β-catenin degradation in an intact Axin1 complex (*Li et al., 2012*) | | | Y | |
| 2013 | Kordes C | The journal of clinical investigation | Hepatic stem cell niches (*Kordes & Häussinger, 2013*) | | | | Y |
| 2014 | Mokkapati S | Cancer research | β-catenin activation in a novel liver progenitor cell type is sufficient to cause hepatocellular carcinoma and hepatoblastoma (*Mokkapati et al., 2014*) | | | | Y |
| 2014 | Oishi N | Liver cancer | Molecular biology of liver cancer stem cells (*Oishi, Yamashita & Kaneko, 2014*) | | | | Y |
| 2014 | Sun G | Current topics in developmental biology | Control of growth during regeneration (*Sun & Irvine, 2014*) | | | | Y |
| 2015 | Jörs S | The journal of clinical investigation | Lineage fate of ductular reactions in liver injury and carcinogenesis (*Jörs et al., 2015*) | | | | Y |

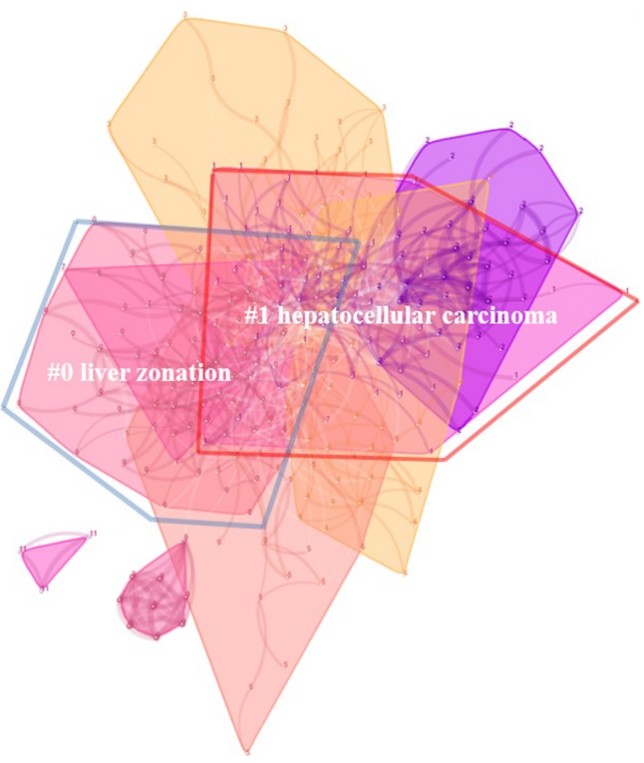

**Figure 4 Two major subtopics, "liver zonation" and "hepatocellular carcinoma" identified during clustering.**

phrases were "liver development," "progenitor cells," "hepatic stellate cells," "liver regeneration," "liver fibrosis," and "epithelial-mesenchymal transition." The bursting phrases after 2008 had the potential to be future hotspots (Fig. 5B).

## DISCUSSION

In the field of Wnt signaling in liver diseases, the number of manuscripts increases annually. Before 2015, the number of studies was even increasing exponentially. After 2016, the annual number of studies was about 180–200. The whole scientific area constituted of 1,768 records, 57,478 references, and a much higher number of relations of citing or being cited. Therefore, although several reviews based on expert opinions have already been published, bibliometric analysis using artificial intelligence algorithms is still needed to provide a bird-view of the whole area, which could also help analyze current trends and predict hotspots in the future.

CiteSpace was used for most of the analyses in this study, and most of its functions were utilized. For instance, to simplify a complex co-citation network of literatures, $g$-index was used as a rule for restriction. However, CiteSpace could also use other restrictions such as the top 50 papers per slice, top 10% per slice, or adjusting the time interval per slice as 2 years (*Chen, 2018*). To measure the importance of each node, four bibliometric indexes were utilized, including total number of citations in co-citation network, centrality, burstness, and usage 2013. The last three indicators were seldom utilized in non-bibliometric publications.

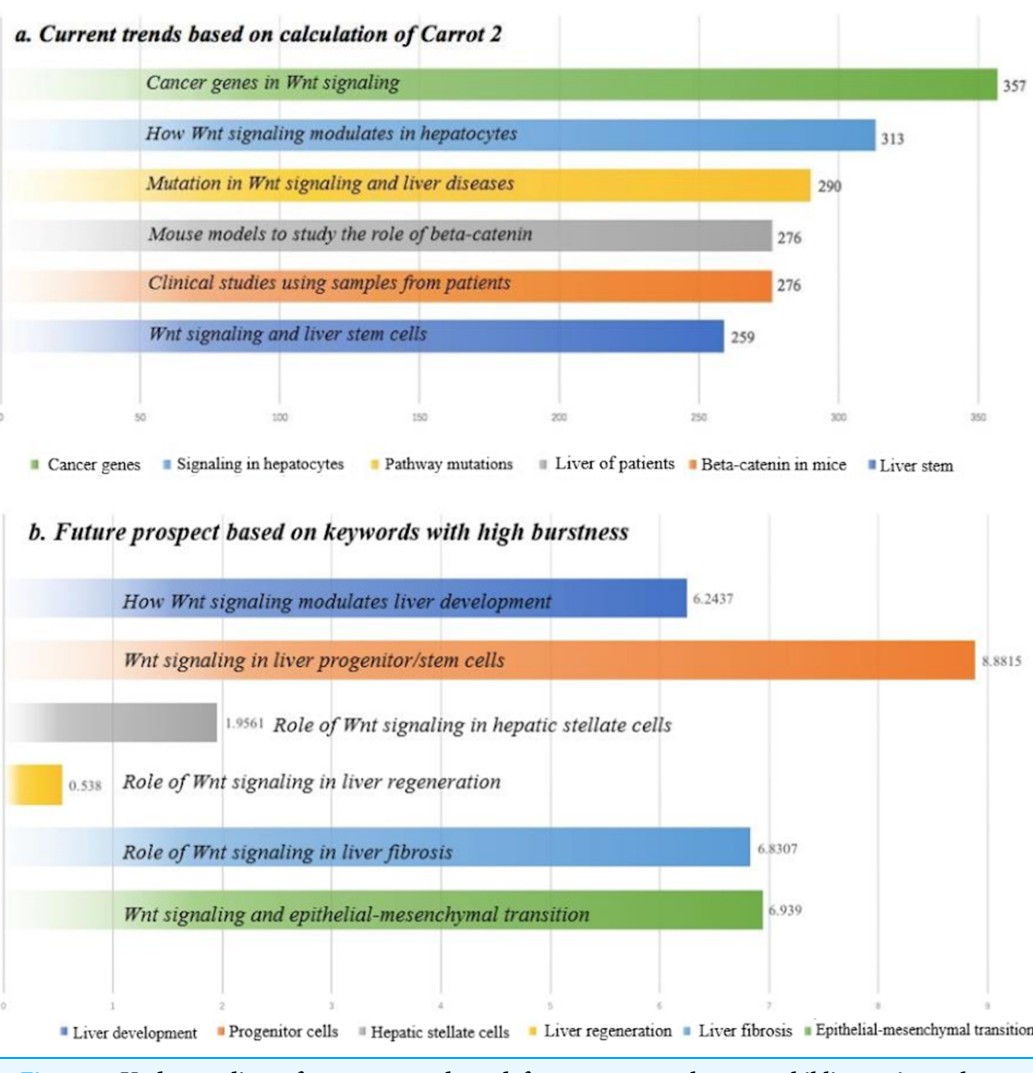

**Figure 5 Understanding of current trends and future prospect bases on bibliometric analyses.**
(A) Current trends based on calculation of Carrot 2. (B) Future prospect based on keywords with high burstness.

Using these indexes, several landmark manuscripts were identified. For example, the work of *He et al. (1998)* was included among the top 5 in centrality and the top 5 in burstness. This manuscript answered a question about the Wnt signaling pathway in the process of carcinogenesis: "How a Growth Control Path Takes a Wrong Turn to Cancer" (*Pennisi, 1998*). This paper was cited by a large number of papers on Wnt signaling in HCC, such as the studies by *Van Nhieu et al. (1999)* and *Merle et al. (2004)*. The work by *Satoh et al. (2000)* was among the top 5 in citation as well as the top 5 in burstness. It suggested that Axin1 might be a molecular target suppressing the growth of HCC. Furthermore, the index of usage 2013 identified manuscripts that were highly valued during these 5 years. For example, the study by *Mokkapati et al. (2014)* suggested that Wnt pathway activation was sufficient for malignant transformation of a type of liver progenitor cells.

By the process of clustering in CiteSpace, the whole science domain was divided into nine subtopics. The largest two clusters are "liver zonation "and "HCC." "Liver zonation"

**Table 2 Summarization of 15 noun phrases in citation network to have burstness.**

| Term | Strength | Begin | End | Sample paper | Journal |
|---|---|---|---|---|---|
| Beta-catenin gene | 8.3092 | 1998 | 2005 | Somatic mutations of the β-catenin gene are frequent in mouse and human hepatocellular carcinomas (*De La Coste et al., 1998*) | Oncogene |
| Nuclear accumulation | 2.9721 | 2002 | 2005 | Nuclear accumulation of mutated beta-catenin in hepatocellular carcinoma is associated with increased cell proliferation (*Van Nhieu et al., 1999*) | The American journal of pathology |
| Adenomatous polyposis coli | 6.4555 | 2002 | 2006 | Oncogenic mutations in adenomatous polyposis coli (Apc) activate mechanistic target of rapamycin complex 1 (mTORC1) in mice and zebrafish (*Valvezan et al., 2014*) | Disease models & mechanism |
| Genetic alterations | 3.7144 | 2002 | 2006 | Altered expression of E-cadherin in hepatocellular carcinoma: correlations with genetic alterations, beta-catenin expression, and clinical features (*Wei et al., 2002*) | Hepatology |
| Beta-catenin mutations | 9.2414 | 2002 | 2005 | P53 gene and Wnt signaling in benign Neoplasms: beta-Catenin mutations in hepatic adenoma but not in focal nodular hyperplasia (*Chen et al., 2002*) | Hepatology |
| Hepatocellular carcinomas | 8.0349 | 2007 | 2011 | Glypican-3 promotes the growth of hepatocellular carcinoma by stimulating canonical Wnt signaling (*Capurro et al., 2005*) | Cancer research |
| Wnt pathway | 2.8812 | 2008 | 2011 | Deciphering the function of canonical Wnt signals in development and disease: conditional loss- and gain-of-function mutations of alpha-catenin in mice (*Grigoryan et al., 2008*) | Genes & development |
| Liver development | 6.2437 | 2008 | 2009 | APC mutant zebrafish uncover a changing temporal requirement for Wnt signaling in liver development (*Goessling et al., 2008*) | Developmental biology |
| Progenitor cells | 8.8815 | 2009 | 2010 | EpCAM-positive hepatocellular carcinoma cells are tumor-initiating cells with stem/progenitor cell features (*Yamashita et al., 2009*) | Gastroenterology |
| Hepatic stellate cells | 1.9561 | 2009 | 2010 | Pregnane X receptor activators inhibit human hepatic stellate cell transdifferentiation in vitro (*Haughton et al., 2006*) | Gastroenterology |
| Aberrant activation | 3.9739 | 2010 | 2011 | Molecular targeted therapies in hepatocellular carcinoma (*Llovet & Bruix, 2008*) | Hepatology |
| Cyclin d1 | 1.0341 | 2012 | 2013 | Immunohistochemical analysis of the progression of flat and papillary preoplastic lesions in intrahepatic cholangiocarcinogenesis in hepatolithiasis (*Itatsu et al., 2007*) | Liver international |
| Liver regeneration | 0.538 | 2015 | 2016 | Beta-Catenin activation promotes liver regeneration after acetaminophen-induced injury (*Apte et al., 2009*) | The American journal of pathology |
| Liver fibrosis | 6.8307 | 2015 | 2016 | Wnt signaling in liver fibrosis: progress, challenges and potential directions (*Miao et al., 2013*) | Biochimie |
| Epithelial-mesenchymal transition | 6.939 | 2016 | 2018 | Noncanonical Frizzled2 pathway regulates epithelial-mesenchymal transition and metastasis (*Gujral et al., 2014*) | Cell |

meant that hepatocytes in different zones of the liver were heterogeneous in biochemical and physiological functions. This difference was also known as metabolic zonation. The Wnt signaling cascade played a dominant role in governing metabolic zonation (*Gebhardt, 2014*), especially in the pathways of glutamine synthesis (*Shigeki et al., 2010*), and drug metabolism (*Burke et al., 2009*). Some papers also highly cited the references in this cluster. For example, one was the review by *Gebhardt & Hovhannisyan (2010)*, which proposed a three-level model for the molecular interpretation of beta-catenin activity in metabolic zonation. In the field of "HCC," dysfunction in Wnt signaling was considered to reactivate some molecular cascades in embryogenesis and induce the

transformation of normal hepatocytes to the malignant phenotype (*Wands & Kim, 2014*). Certain molecules in the Wnt pathway, including the extracellular, cytosolic, and nuclear participants, were explored to determine whether they were potential targets for therapeutic interventions (*Pez et al., 2013*). The liver stem cell pathway was also mentioned in cluster "stem cell niche" and "liver stem cell." Wnt signaling is essential in stem cell control, as a proliferative and self-renewal signal. Several high-quality reviews have analyzed this topic (*Clevers, Loh & Nusse, 2014*; *Reya & Clevers, 2005*). Finally, it was noteworthy to find that cluster "human colorectal cancer" was not directly related to liver research. However, since some important Wnt molecules were firstly investigated in the study of colon cancer, references of this subtopic were also highly cited by studies focusing on liver research.

Bibliometric methods can quantitatively identify some active words in research, and then judge the emerging trends and future prospects in a science domain. These noun phrases could also be directly derived from author keywords or the "key words plus" generated by WOS. However, in this study, we used an artificial intelligence-based "Part-of-Speech" technique to generate noun phrases directly from the titles/abstract, and then constructed a co-citation network of scientific ideas with more importance. This novel strategy was not feasible with traditional reviews. Through this strategy, we found some important phrases after 2008. For example, the phrase "liver fibrosis" had its bursting period during 2015 and 2016. One study using this phrase was the review by *Miao et al. (2013)*, in which Wnt signaling was summarized to promote liver fibrosis by enhancing hepatic stellate cell activation and survival. The phrase "epithelial-mesenchymal transition" had its bursting period during 2016 and 2018. The study by *Gujral et al. (2014)* investigated the Wnt receptor Frizzled2 (Fzd2) in epithelial-mesenchymal transition, and suggested that using an antibody to Fzd2 was a novel way to inhibit tumor growth and metastasis. Together with the results from Carrot 2 and "Part-of-Speech" analysis, understanding of current trends and future prospects were interpreted in Fig. 5 based on bibliometric analysis.

This study had several limitations. One was that the co-citation analysis of manuscripts was only feasible in papers derived from the WOSCC. Therefore, searching results from other databases including PubMed, Ovid, Scopus, and Google Scholar was not feasible using CiteSpace. Secondly, this analysis did not show clusters about Wnt signaling in alcoholic liver diseases or nonalcoholic steatohepatitis. Although there were several studies on Wnt signaling in these diseases (*Huang et al., 2015*; *Teratani et al., 2018*), the number of studies highly cited was not enough. Therefore, the simplified co-citation map did show these papers as important nodes.

In conclusion, using the bibliometric methods, we quantitatively summarized the advancements and emerging trends in Wnt signaling in liver diseases. These bibliometric findings may pioneer the future direction of this field in the next few years, and further studies are needed.

## ABBREVIATIONS

**WOSCC**    Web of Science Core Collection
**HCC**    hepatocellular carcinoma

| WOS | Web of Science |
| INSERM | Institut National de la Santé et de la Recherche Médicale |
| Fzd2 | Frizzled2 |

### Funding
This research was supported by Zhejiang Provincial Natural Science Foundation of China under Grant No. LQ17H030003. The funders had no role in study design, data collection and analysis, decision to publish, or preparation of the manuscript.

### Grant Disclosures
The following grant information was disclosed by the authors:
Zhejiang Provincial Natural Science Foundation of China under: LQ17H030003.

### Competing Interests
The authors declare that they have no competing interests.

### Author Contributions

- Guangyi Jiang conceived and designed the experiments, performed the experiments, contributed reagents/materials/analysis tools, prepared figures and/or tables, authored or reviewed drafts of the paper, approved the final draft.
- Chiung-Kuei Huang analyzed the data, prepared figures and/or tables.
- Xinjie Zhang analyzed the data, prepared figures and/or tables.
- Xingyu Lv analyzed the data, prepared figures and/or tables.
- Yifan Wang analyzed the data, prepared figures and/or tables.
- Tunan Yu conceived and designed the experiments, performed the experiments, contributed reagents/materials/analysis tools, authored or reviewed drafts of the paper, approved the final draft.
- Xiujun Cai conceived and designed the experiments, performed the experiments, authored or reviewed drafts of the paper, approved the final draft.

### Data Availability
  The raw measurements are available in the Supplemental Files.

### Supplemental Information
Supplemental information for this article can be found online at http://dx.doi.org/10.7717/peerj.7073#supplemental-information.

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
