# Peer review of "Wnt signaling in liver disease: emerging trends from a bibliometric perspective"

_PeerJ, doi:10.7717/peerj.7073_

## Round 0.1 · original submission · Major Revisions

The reviewers have raised several major questions which should be addressed in the revised manuscript.

Reviewer 1 ·

Basic reporting

N/A

Experimental design

N/A

Validity of the findings

N/A

Additional comments

I thank the editor for providing an opportunity to review the article “Wnt signaling in liver disease: emerging trends from a bibliometric perspective”.
Since this is not an original research article, it can be considered for “resource” or “prospective” group of publication.
The article is written well and can be considered for publication after addressing these comments:


1. Why Wnt signaling was specifically chosen for bibliometric analysis? What was the motivation behind?
2. Authors should include flow diagram for the Wnt signaling pathway, to give better prospective for readers.
3. Figures should be merged, and the number should be reduced (there are figures for each point). Fig 1 should be about summary of past research and Fig 2 should be future prospective research topics.
4. Take home message for this article is to give current trends and future prospects to Investigators about Wnt signaling. This point should be highlighted and included in the figure, rather than summarized in the table.
5. For example, Fig 6 should be about what could be potential future research topics rather than what keyword used most frequently. Authors should include a figure to speculate future research topics and importance based on the bibliographic analysis. And also discuss why authors think that these suggested topics would be important to address in the future for Wnt signaling research.

Reviewer 2 ·

Basic reporting

The manuscript entitled Wnt signaling in liver disease: emerging trends from a bibliometrics perspective by Guangyi Jiang et al. summarized the advancements and emerging trends in Wnt signaling in liver diseases using bibliometrics methods. Although this is theoretical paper and this type of analysis had already been performed earlier and many in social science research yet it is new for biological science field and comprehensive overview for field development.

Experimental design

NOT APPLICABLE

Validity of the findings

Limited impact.

Additional comments

There are few major concern that I mentioned below:

In result section:
Line 108: The subheading is “Authors, institutions, and nations”. But the authors started the paragraph with manuscript numbers. There is no analysis on “author” been presented under this subheading. I would suggest that author need to work on result. Author should structure their result section in a systematic way. For example, author may consider the following subheadings to structure their result:

1 Publication outputs/Numbers
2 Country/Geographical distribution
3 Institution analysis
4 Research hotspots
5 Journal analysis/Landmark manuscript
6 Research areas/ subtopics
7 Author analysis
8 Research trends/Evolution of scientific ideas

Line 128: “3. Important subtopics” Important: is an adjective which may be refrained from using in subheading.

Figures:
Figure figure number are not matching. Figure 3 is Fig 2b. Author need to rewrite the all the figure legends.

Figure 1: The numbers on x and y axis are hardly visible. Author could write number on top of each year’s bar as well or may present by line diagram instead of bar.

Figure 2a/2b: The figure doesn’t represent the text in result section. In text, Line 114-116 author has mentioned the individual countries publication which is not apparent in figure. I would suggest all the countries with their publication in table form. In figure 2b “Peoples R China” can be written as “China” similar to USA for United State of America.

Figure 3a/4: Map can show the distribution of institute on globe, it’s hard to predict the institution., again table will work better than globe.

Figure 3b/5: It is hard to read institute name and correlate with map in figure. This data can be presented in simpler form like horizontal bar graph.

Figure 4a/6: I would suggest removing or change this figure with as nothing is clear in picture. Please elaborate the every detail of figure in its legend e.g. what does lines in figure signify, what are nodes and color of nodes signify?

Figure 4b and figure 5 is not clear at all. Texts in figures are unreadable, neither it is clear that which node or which part of the figure is highlighted by text.

Figure 6/9: In Foam tree, all the keyword has same area, which obscure the purpose of figure.

Remove language certificate from supplementary file.

---

## Round 0.2 · accepted · Accept

The manuscript is ready for publication. Please improve the quality of table.

Reviewer 1 ·

Basic reporting

N/A

Experimental design

N/A

Validity of the findings

N/A

Additional comments

Authors have included all suggestions and addressed all comments on the manuscript. This article can be accepted for publication.

Reviewer 2 ·

Basic reporting

The manuscript entitled Wnt signaling in liver disease: emerging trends from a bibliometric perspective by Guangyi Jiang et al. summarized the advancements and emerging trends in Wnt signaling in liver diseases using bibliometric methods.

Experimental design

Not Applicable

Validity of the findings

Not Applicable

Additional comments

Manuscript has improved significantly after revision.
Few minor concerns listed below:
Author should check the journal’s guideline to adjust the ‘Font size’ of the manuscript, especially in ‘Tables’. It is hard to visualize the text of tables.

Surprisingly ‘Acknowledgement’ is ‘Not Applicable’??!!